# Possible Stress–Neuroendocrine System–Psychological Symptoms Relationship in Pregnant Women during the COVID-19 Pandemic

**DOI:** 10.3390/ijerph191811497

**Published:** 2022-09-13

**Authors:** Giulia Gizzi, Claudia Mazzeschi, Elisa Delvecchio, Tommaso Beccari, Elisabetta Albi

**Affiliations:** 1Department of Philosophy, Social Sciences and Education, University of Perugia, 06123 Perugia, Italy; 2Department of Pharmaceutical Sciences, University of Perugia, 06126 Perugia, Italy

**Keywords:** COVID-19, pregnant women, confinement, anxiety, depression, mental disorders

## Abstract

The COVID-19 pandemic induced long-term damages that weigh on the national health systems of various countries in terms of support and care. This review aimed to highlight the mental health impact of the COVID-19 pandemic in pregnant women. We first report data on the immune system physiopathology and the main viral infections in pregnancy, including COVID-19. Then, the attention is focused on the main factors that affect the mental health of pregnant women during the COVID-19 pandemic, such as (1) the fear of being infected and transmitting the infection to the fetus, (2) the cancellation of checkups and pre-child courses, and (3) confinement and the inability to have close friends or a partner at the time of delivery or in the first days after delivery, as well as family tensions. Because of all this, pregnant women find themselves in a stressful condition independent of the pregnancy, and thus experience anxiety, depression, insomnia, hostility, delirium, and an alteration of the mother–baby relationship. Several studies have shown an involvement of the hypothalamic–pituitary–adrenal axis and the hypothalamic–pituitary–thyroid axis in response to the pandemic. We propose a possible involvement of the neuroendocrine system as a mediator of the psychological symptoms of pregnant women induced by COVID-19-related stress.

## 1. Introduction

Pregnancy is generally considered a critical period for women. Pregnancy is associated with dramatic changes in metabolism, hormone production, mood, and the immune system. Many studies report that different changes are closely linked to each other and depend on circadian rhythms [1]. The hormonal balance is regulated by the brain in the hypothalamus–pituitary axis. The main hormonal actor in pregnancy is progesterone, which allows implantation of the fertilized egg and silences uterine contractions during the pregnancy period [2]. During late pregnancy, a reduction in progesterone contributes to the progression of parturition and the initiation of lactation [2]. Progesterone influences the cellular metabolism [3] and mood symptoms [4]. Moreover, during pregnancy, estradiol is produced by the placenta, inducing various effects on the immune system and the brain [5]. Notably, estradiol in pregnancy is associated with reduced gray matter volume in the left putamen, with a reduction in cognitive performance and an increase in negative affect symptoms [6]. Additionally, during late pregnancy, prolactin production is responsible for the low anxiety levels in lactating women [7], although anxiety can be experienced in women who have difficulty breastfeeding.

In general, hormonal modifications allow the pregnancy to proceed in a good state of physical–psychological balance, and maternal well-being also includes having a positive childbirth experience [8].

This review was intended to highlight how women, in a state of hormonal and immune changes that occur during pregnancy, react to a viral infection. It was also highlighted how a pandemic has different implications for pregnant women compared to a simple viral infection. In fact, during the COVID-19 pandemic, concern for the aggressiveness of the virus and its possible maternal–fetal transmission, hospital organization problems, confinement, and alteration of interpersonal relationships induced the appearance of psychological symptoms such as anxiety, depression, insomnia, post-traumatic stress, negative affect, and delirium in pregnant women. The possibility that the neuroendocrine system acts as a bridge between the viral infection/stress induced by COVID-19 and the consequent psychological symptoms has been considered.

Certainly, pregnancy and childbirth are joyful experiences for a woman. However, many women have symptoms of depression and anxiety during pregnancy, which are risk factors for postnatal depression. Generally, hormonal changes contribute to emotional well-being, but different factors during pregnancy or the experience of childbirth can impact a woman’s mental health. In particular, the moment of childbirth, triggered by hormonal changes, is a critical phase as it depends on various variables such as health workers, hospital, support available for the woman, previous childbirth experiences, and more.

## 2. Viral Infection and Immune System in Pregnancy

Pregnancy creates a particular and unique immunological condition. The maternal immune system must allow the tolerance of the fetus and at the same time protect against micro-organisms. During early trophoblast invasion and at parturition, a pro-inflammatory environment predominates, while in the second and third trimesters, an anti-inflammatory environment is created [9]. The balance between innate and adaptive immunity shifts in favor of innate mechanisms that include the action of natural killer cells, lymphocytes, mast cells, eosinophils, basophils, macrophages, neutrophils, and dendritic cells, and mediate the induction of tolerance and defense against infection [10].

Innate mechanisms are activated early (first 5–7 days of infection) and may be sufficient to eliminate the virus or to limit its replication by reducing its dissemination [11]. Therefore, in pregnancy, innate immune cells and effector mechanisms are upregulated [9]. Interestingly, a Th1 inflammation-like condition is found early in pregnancy, with the shift to a temporal Th2-biased immune tolerance state during the second trimester; a second shift occurs later [12]. However, a viral infection in pregnancy might be a risk factor for obstetric complications [13], maternal morbidity and/or mortality, and vertical transmission to the fetus [14].

The most troubling forms of non-obstetric viral infections in pregnancy are pneumonia. In the past twenty years, three viruses, all belonging to the coronavirus family affecting the respiratory system, have been believed to be responsible for respiratory problems in pregnant women: severe acute respiratory syndrome coronavirus (SARS-CoV), Middle East respiratory syndrome coronavirus (MERS-CoV), and COVID-19 (SARS-CoV-2) [15]. Today, we can say that the spread of COVID-19 has largely exceeded that of SARS and MERS, albeit with a lower mortality rate. The data reported by different studies on the effect of COVID-19 in pregnancy are conflicting. Consequences of COVID-19 infection on implantation, fetal growth and development, and newborn and mother damages have yet to be determined. Although pregnancy is characterized by an immunocompromised and hypercoagulable state, evidence indicates that the contribution of COVID-19 is not very significant in maternal morbidity and mortality, and no vertical transmission of the virus has been reported yet [16]. The principles of medical behavior towards a COVID-19-positive pregnant woman include early isolation, tests for co-infections, oxygen therapy, control of body fluid, prevention of thromboembolism, monitoring of fetal and uterine contraction, and planned delivery. If the condition of the woman worsens, early mechanical ventilation for progressive respiratory failure and an evaluation for a premature birth are important.

Knight et al. [17] showed in a population of 427 pregnant women hospitalized with COVID-19 in the Oxford (U.K.) no evidence of an increased risk of severe disease compared to the general hospital population. A case of a 37-year-old French woman diagnosed with COVID-19 after the onset of postpartum fever, with mild blood picture abnormalities, has been described. After 5 months, the woman had COVID-19 reinfection with a serious clinical picture, indicating that outside the pregnancy period, the infection was more prominent [18]. Th1/Th2 shifts are responsible for the risk of developing severe complications in patients with COVID-19 [19]. Of note, these shifts might significantly affect the onset of depression symptoms or even lead to them [20]. In a study conducted in China, women who gave birth before the COVID-19 pandemic and women exposed to COVID-19 during pregnancy were considered. Women in the exposure group were more susceptible to hypertension during pregnancy and were more likely to have premature babies [21]. It should also be considered that percentage of cesarean delivery was increased [22]. The authors commented that the increase in cesarean delivery in the COVID-19 period could be due to the preference of private hospitals. This could justify the greater increase in the Mediterranean regions compared to the Aegean, Western Marmara, and Black Sea regions due to the greater economic wealth in the Mediterranean regions. Moreover, breastfeeding was reduced in women exposed to COVID-19 compared to control women for maternal–neonatal separation that was enforced to limit the spread of the virus [23]. Thus, at the moment, the studies on the physical consequences, in terms of direct damage and complications induced by COVID-19 in pregnant women, are conflicting.

## 3. The Psychological Impact of the COVID-19 in Pregnancy

A peaceful pregnancy and a positive birth experience are essential for maternal well-being and a good mother–child relationship [24]. A difficult pregnancy and/or birth can lead to the onset of postpartum depression that typically occurs after the birth of the child until the end of the first year of the infant’s life [25]. Lebel et al. [26] reported that the symptoms of anxiety and depression were found in 10–25% of pregnant women in non-clinical conditions. All psychological symptoms of pregnant women were exacerbated by the COVID-19 pandemic, as well as additional symptoms that have emerged.

An anonymous online survey of pregnant or postpartum women, international in scope, carried out in 12 languages, showed that the levels of post-traumatic stress and anxiety/depression were very high and loneliness was highly prevalent [27]. An online study conducted at the University of Michigan, 3–24 April 2020, among 2740 pregnant women from 47 different states showed that the anxiety that manifested in pregnant women during the period of COVID-19 did not depend on the pregnancy itself [28]. Goyal and Selix showed that COVID-19 induced stress in pregnant women by increasing their susceptibility to perinatal anxiety and mood disorders (PAMD) [29]. The authors concluded that it would be necessary to develop national policies aimed at prenatal and postpartum economic and psychological assistance. This observation was very important because it has been shown that socio-economic conditions influence the psychological response of pregnant women to COVID-19. Berthelot et al. [30] reported that women with a previous psychiatric diagnosis or with a low socioeconomic status would be more prone to high stress and psychiatric symptoms in pregnancy. Mei et al. [31] stated that a high level of education reduced the risks of depression, anxiety, and stress in pregnant women compared to a low level of education. However, an interesting study conducted at Mount Sinai Hospital in New York City showed that pregnant women living in a state of poverty benefited from social restrictions [32]. This could have been due to working from home, which overcame the problem of the lack of flexibility of hours due to formal and informal work obligations after the birth of the child. Smart working allowed the future mother to be able to tend to the baby without added costs for assistance during work [32].

### 3.1. Stressful Conditions

Recent studies have highlighted the negative impact of the COVID-19 pandemic on maternal mental health for stressful conditions linked to (1) virus: the fear of contracting the infection and passing it on to the child, as well as the fear of a problematic birth; (2) hospital: the cancellation of check-ups and pre-birth courses; (3) physical–psychological isolation: confinement accompanied by a lack of family and friend support, increased interpersonal tensions, and concern about the unavailability of care (Figure 1).

The first point had an impact on the mental health of the mothers especially because of they were going through a pregnancy experience or a childbirth experience that did not correspond to their expectations. In fact, in Italy (Udine hospital), a study was conducted on 258 pregnant women with the aim to investigate the reaction to the COVID-19 pandemic by two questions: the first was intended to know if the woman had knowledge about COVID-19, and the second was if the woman knew how to control the possibility of getting infected. Women were aware of COVID-19 and the related problems [33]. In a study conducted in Canada among 1987 pregnant women, the reported symptoms were in part due to concerns with the risk of contracting the virus and its effects on the health of the mother and baby [26]. Similar concerns emerged from a multi-center and cross-sectional study involving 12 provinces of China in which the possibility of contracting the infection and consequently creating problems for oneself and the baby was a prominent concern in many pregnant women [34].

The second point was strictly dependent on the reorganization of the national health system for the pandemic emergency. Part of the hospitals were dedicated to COVID-19 patients, as were doctors and nurses. The consequence was the reduction in follow-up visits and interventions for non-COVID-19 patients. Non-urgent checkups and preparatory courses were canceled for pregnant women to minimize the risk of contagion. Again, pregnant women from different parts of the world were concerned. Italian pregnant women were worried about the cancellation of medical appointments [35]. In March 2020 in Canada, the provinces imposed restrictions with consequent hospital reorganizations. As a result, 4604 English- and French-speaking pregnant Canadians perceived a lack of clinical support [36]. This psychological condition was confirmed by another Canadian study involving 1987 pregnant women who raised concerns about not receiving adequate care [26].

Likewise, in completely different territory, pregnant Chinese women declared a relevant concern if they delayed follow-up visits [34].

The third point opened the door to a general problem concerning all the circumstances in which individuals found themselves in a condition of confinement. Solitary confinement has been found to have a variety of negative effects dependent on the duration and conditions [37]. A Spanish study conducted in Valencia on 97 pregnant women found that confinement inevitably led to a reduction in physical activity and an increase in sedentary lifestyle, with consequent repercussions on the mood of expectant mothers [38]. This condition was aggravated by the difficulty of interpersonal relationships due to confinement, as well as by family tensions due to the change in lifestyle and, again, by the inability of the partner to attend the birth or be with the future mother in the first days after the birth. The difficulty of interpersonal relationships, studied by the Social Support Effectiveness Questionnaire (SSEQ), was highlighted by the Canadian study reported above [26]. The importance in Canada of social support for pregnant women during the pandemic was also shown by Khoury et al. [39].

A condition of suffering due to physical absence and/or the absence of support from one’s partner in the perinatal period was detected in an Italian study conducted on 575 pregnant women [40]. Moreover, 114 pregnant English-speaking American women, residents of Sedgwick County, adopted strategies for self-care, to avoid the risk of contamination, and for a good family organization [41]. The authors reported that a quarter of patients complained of a decrease in support from family, friends, coworkers, and support services. Additionally, negative behaviors attributed to the pandemic were reported, such as a decrease in physical activity and an increase in tobacco and alcohol consumption [41]. In a Chinese study, pregnant women recruited before and after the COVID-19 pandemic were questioned about their families via the Family Environment Scale (FES). From the analysis of the data, reduced family cohesion and an increased level of conflict were highlighted [42].

### 3.2. Psychological Symptoms

Interestingly, from European and extra-European studies emerged many symptoms triggered by one or more of the above causes (Figure 2). The studies were conducted either by analyzing the symptoms that appeared during the COVID-19 period or by comparing the symptoms between the COVID-19 period and the non-COVID-19 period.

#### 3.2.1. COVID-19 Period

In order to understand the effects of the COVID-19 pandemic on the mental health of pregnant women, numerous studies have been conducted in different countries around the world. Most of them concern European, American, and Asian countries. The main causes that led to mental disorders were the decrease in the perception of general support, family economic difficulties and the possible state of unemployment, a tendency to disobey the rules of isolation, and a low level of education [43].

##### European Countries

In Italy, the impaired mental health linked to the COVID-19 pandemic has been demonstrated in several studies. Two of these were carried out at Udine hospital, demonstrating an association between the COVID-19 pandemic and clinical symptoms of anxiety and depression [33,44] and obsessive compulsive disorder [45]. In the first study, 258 pregnant women were analyzed by the General Anxiety Disorder-7 (GAD-7), the Patient Health Questionnaire-2 (PHQ-2), and an obsessive compulsive disorder (OCD) screening [33]. In the second study, 232 pregnant women were analyzed by the Italian version of the Pandemic-Related Pregnancy Stress Scale (PREPS), the Revised Prenatal Distress Questionnaire (NuPDQ), the General Anxiety Disorder-7 (GAD-7), and the Patient Health Questionnaire-2 (PHQ-2) [44].

The high level of depressive symptoms was confirmed by an online anonymous survey prepared by the University of Roma (Central Italy) administered for 6–8 weeks to 286 women who had given birth during the COVID-19 pandemic [45]. The results showed that 64 women had SARS-CoV-2 infection during pregnancy and consequently were separated from the newborn, with low probability of breastfeeding. The Edinburg Postnatal Depression Scale showed that postnatal depression was relatively common in the whole cohort, but the likelihood of being affected by this mental disorder was higher in women with COVID-19 [45]. The fact that the COVID-19 period was demanding and stressful for pregnant women, with a significant impact on their well-being, was confirmed by another online study. It was organized by the University of Milan and performed with 575 pregnant Italian women by the administration of self-assessment questionnaires concerning the presence of anxiety disorders, depressive symptoms, and post-traumatic stress disorder [40]. The possibility that anxiety in pregnant women during the COVID-19 period could negatively affect prenatal attachment was demonstrated with an online study conducted by a group of researchers from the University of Cosenza (Southern Italy) with self-assessment questionnaires aimed at analyzing socio-demographic and obstetric characteristics, psychological distress (form STAI Y-1-2 and BDI-II), and prenatal attachment (PAI) [46].

In other European countries, the effects of COVID-19 on the mental health of pregnant women were not very different from those found in Italy.

In Spain, it was demonstrated by the Symptom Checklist-90 Revised (SCL-90-R) that for 131 pregnant women, stress and insomnia were predictive variables of anxiety and depressive symptoms related to COVID-19 [47]. In France, a relevant state of anxiety/depression during pregnancy in the COVID-19 period was highlighted by a socio-demographic questionnaire, the Spielberger Trait Anxiety Inventory, the Edinburgh Postnatal Depression Scale, and the Multidimensional Scale of Perceived Social Support [48]. Moreover, postpartum anxiety and depression were evaluated with the City Birth Trauma Scale, the Interpersonal Emotional Regulation Questionnaire, and the Posttraumatic Growth Inventory [48].

##### American Countries

The COVID-19 pandemic has also affected the mental health of pregnant women in America. In a Canadian study (North America) performed with the Edinburgh Depression Scale (EPDS), it was shown that more than half of pregnant women during the COVID-19 period had clinically relevant depression symptoms, and more than a third had anxiety symptoms [28]. In Colombia (Southern America), a study that involved seven cities indicated the psychological consequences of the pandemic as anxiety, insomnia, and depression in more pregnant women than those who had actually been affected by the virus [49].

##### Asian Countries 

The situation was no different in Asia. Of note, in China, a depressive state that affected more than a quarter of pregnant women during the COVID-19 period was shown by the Patient Health Questionnaire-9 (PHQ-9) [34].

#### 3.2.2. COVID-19 Period Versus Non-COVID-19 Period

Several studies have been carried out with the aim of comparing the mental conditions of pregnant women during the COVID-19 period compared to the non-COVID-19 period.

##### Differences in Countries

It is known that stressful conditions can be encountered in women during pregnancy for different causes, such as ethnicity, socio-economic status, marital status, employment activity, and cultural level, as well as relationships with parents, partners, and friends [46], with consequences on the health of the child. In fact, it has been reported that stressful maternal conditions can induce the possible onset of emotional problems, cognitive disorders, attention deficit, and hyperactivity in the child [50]. Prenatal stress was particularly high in twin pregnancies. This involved approximately 44% of women in early pregnancy and 51% in late pregnancy, indicating that the stressful condition increased as pregnancy progressed. In the study, it was demonstrated that the premature rupture of the membranes was attributable to the condition of terminal stress. Interestingly, the stressful condition correlated with the women’s BMI and education level [51]. Therefore, numerous studies have been conducted to highlight the difference in the mental health of pregnant women in the non-COVID-19 period and the COVID-19 period. Certainly, demographic factors influence the mental condition of pregnant women both in the non-COVID-19 period and the COVID-19 period [44,52]. However, problems related to the pandemic such as job difficulties resulting in low income, especially for single women, as well as the loss of relatives made the situation much more difficult for pregnant women during the COVID-19 period [53].

##### European Countries

In Italy, the link between the high levels of anxiety, depression, and hostility of women in the third trimester of pregnancy and the COVID-19 pandemic was demonstrated with the Profile of Mood States and the Multidimensional Scale of Perceived Social Support by comparing the results with those of pregnant women in the non-COVID-19 period [54].

##### American Countries

The differences in mental health in pregnant women during the non-COVID-19 period and the COVID-19 period have also been demonstrated in America. To assess the differences between two groups of Boston women in relation to childbirth stress, maternal bonding, and breastfeeding, 1611 women who gave birth during the pandemic and 640 women who gave birth in the non-COVID-19 period were studied with an anonymous internet survey. The results showed that mothers exposed to COVID-19 were found to have a much more acute stress response with symptoms of post-traumatic stress disorder related to childbirth and breastfeeding problems with subsequent mother–baby bond problems compared to mothers in the non-COVID-19 period [55]. The differences between the two periods were also revealed with the Cambridge Concern Scale (CWS), the Center for Epidemiologic Studies Depression Scale (CES-D), the Insomnia Severity Index (ISI), and the Multidimensional Perceived Social Support Scale (MSPSS). Moreover, among 303 pregnant women from Ontario, Canada, more than half of the patients had a high level of depression, more than a third had a high level of worry, and more than a fifth had a high level of insomnia during the COVID-19 period in comparison with the non-COVID-19 period [39]. Additionally, higher levels of anxiety, depression, dissociation, post-traumatic stress disorder, and negative affect symptoms were demonstrated in pregnant women during the COVID-19 pandemic compared to the non-COVID-19 period in Quebec, Canada, evaluated with the Kessler Distress Scale (K10), the Post-traumatic Checklist for DSM-5 (PCL-5), Dissociative Experiences Scale (DES-II), and the Positive and Negative Affect Schedule (PANAS) [30].

##### Asian Countries

The comparison of the mental conditions of pregnant women between the COVID-19 period and the non-COVID-19 period was also carried out in Asia. At Wuhan Children’s Hospital of China, the data obtained from the Novel Coronavirus-Pregnancy Cohort (NCP) study that included 531 participants and the Healthy Baby Cohort (HBC) study that included 2352 participants were compared. Women analyzed during the pandemic had a higher level of depression but a lower risk of stress than non-COVID-19 women [31]. The authors commented that the stress reduction could be related to the fact that the women participating in the study did not have to travel to the workplace, reducing work-related stress. An increased level of depression but also of somatization, anxiety, hostility, and sleep disturbances during the COVID-19 period compared to the non-COVID-19 period was demonstrated in a Chinese study conducted at Zhejiang Hospital and Shanghai University using the Symptom Checklist-90 Revised (SCL90-R) and Pittsburgh Sleep Quality Index (PSQI) questionnaires [41].

In addition to all the symptoms listed above, a case of delirium was found at a Mumbai hospital (India) in a woman in the 30th week of pregnancy affected by COVID-19 with already diagnosed anemia and preeclampsia. A premature birth followed, and the episode occurred again 4 days after delivery. Symptoms subsequently improved and the patient was discharged [56].

## 4. Possible Involvement of Hypothalamic–Pituitary–Adrenal Axis and Hypothalamic–Pituitary–Thyroid Axis in the Mental Health Impact of COVID-19

As above reported, the fear of the virus, of the deficiency of health organizations, and of social isolation induced a stressful condition in pregnant women with consequent anxiety, depression, insomnia, post-traumatic stress, negative affect, and delirium. It has been reported that depression might be associated with changes in the hypothalamic–pituitary–adrenal axis (HPA) and the hypothalamic–pituitary–thyroid axis (HPT) [57]. Of note, the stress acted on HPA with a consequent release of cortisol into the blood [58]. It was demonstrated that fluctuations in cortisol levels were often associated with mental disorders [59]. Thus, cortisol was considered one important biomarker of anxiety and depression disorders [60]. The high cortisol level in the blood together with anxiety and depression symptoms have been related to psychosocial stressors [61]. Interestingly, the effect of maternal elevations of blood cortisol due to prenatal stress had short- and/or long-term effects on the fetus, dependent on the length of exposure [62]. As reported above, the COVID-19 pandemic induced psychosocial stressors that particularly affected pregnant women.

Moreover, it has been demonstrated that confinement-related human damages were due to a variation in the functionality of the thyroid gland [63]. Notoriously, this gland controls the metabolism of numerous systems and apparatuses, including the immune, cardiovascular, gastrointestinal, and nervous systems [64]. In adults, hypothyroidism impaired mental health; of note, 20% of patients were affected by depression [65]. Hua et al. [66] performed a study in Shanghai (China) on the relation between home confinement forced by COVID-19 in pregnant women and the thyroid function, showing that confinement made women more vulnerable to stressful agents, resulting in hypothyroidism. This is relevant considering the relation between the thyroid and mental health during the COVID-19 pandemic [67]. Certainly, the direct relationship of confinement, thyroid alteration, and mental symptoms has not always been studied, and the possibility that mental confinement symptoms are independent of the thyroid cannot be excluded. Beyond the thyroid-dependent or thyroid-independent mechanism of action, confinement has certainly been shown to create humoral problems for pregnant women.

Thus, adrenal and thyroid axes were implicated in the stress induced by the COVID-19 pandemic. It was believed that the implication mechanism required the activation of the immune system. In fact, at the beginning of a viral infection, proinflammatory cytokines are released; they cross the blood–brain barrier and activate the neuroendocrine system [68]. Although the changes in the neuroendocrine system in COVID-19 have not been fully elucidated, it has been shown that deregulation of the brain–peripheral hormonal axes occurs, with hyperactivation or hypoactivation conditions. Hyperactivation is believed to result from the proinflammatory cytokine storm, while hypoactivation could be dependent specifically on tumor necrosis factor alpha (TNF-α) and transforming growth factor beta (TGF-β) [69]. Considering that the HPA and HPT respond to stress and that their changes in regulation affect mental health, it is possible to hypothesize a change in the immuno-emotional regulatory system with the involvement of the two hormonal axes in the psychological symptoms that pregnant women show in response to virus infection and stress conditions induced by the COVID-19 pandemic (Figure 3). At the moment, there are no data demonstrating this direct relationship, but it is possible to suppose it by cross-referencing the results of the various studies reported in the literature. Future studies might be oriented in this direction.

## 5. Study Limitations

Some limitations of the study need to be considered. We had no information on the long-term impact of maternal stress on babies that could be relevant for future studies conducted after the pandemic. In addition, we reported studies conducted on hospitalized pregnant women; data on women who have undergone a home birth are lacking. Moreover, our main objective was not to explore the mortality rates but rather the mental health of pregnant women during the COVID-19 pandemic. Thus, we did not obtain mortality data.

## 6. Discussion and Future Perspectives 

The COVID-19 pandemic remains one of the main stressors in pregnancy and childbirth for women both in Europe and in non-European countries. New and unexpected personal, social, health, and economic conditions have an impact on women’s mental health.

From the research reported in this review, it is clear that a virus-induced pandemic creates a series of problems in pregnant women that go far beyond a simple viral infection. Concern about the physical effects of the COVID-19 virus on pregnant women and babies, regarding possible maternal–fetal transmission, is similar only in part to that of other aggressive viruses. However, since the long-term effects are still not well defined, due to the wide variety of symptoms and the short time that has elapsed since the beginning of the pandemic, COVID-19 is even more worrying. Furthermore, the organizational problems that inevitably hit all hospitals and the consequences of confinement have strongly affected the mental balance of pregnant women. It is too early to study the possible cognitive–behavioral repercussions of children born to pregnant mothers who have been affected by COVID-19. Certainly, in severe cases of maternal infection, the baby does not have contact with the mother and is not breastfed, with probable consequences. Multicenter, prospective cohort studies with adjustments for known confounding factors are needed to explore the impact of infection control policies on the mother–infant relationship. It is important to implement policies aimed at supporting early and safe breastfeeding practices, with innovative information aimed at tackling a pandemic.

## 7. Conclusions

Anxiety, depression, insomnia, post-traumatic stress, and negative affect could have long-term effects, with repercussions on the woman herself and on the mother–child relationship which, in addition to the problems of health organizations, weighs on the national health system in terms of assistance and care. It would therefore be advisable to invest in adopting useful measures to prevent pregnant women’s discomfort induced by the pandemic, e.g., by improving the health organization and creating specific internet sites where women can come into direct contact with specialized personnel. Psychologists who can support the general practitioner could be useful to help future mothers in their relationship with their child, family, friends, and co-workers so that women can experience a peaceful postnatal period. The possible involvement of the hypothalamic–pituitary–adrenal axis and hypothalamic–pituitary–thyroid axis in the mental health impact of COVID-19 is intended to be an idea that can serve as a stimulus for future studies.

## Figures and Tables

**Figure 1 ijerph-19-11497-f001:**
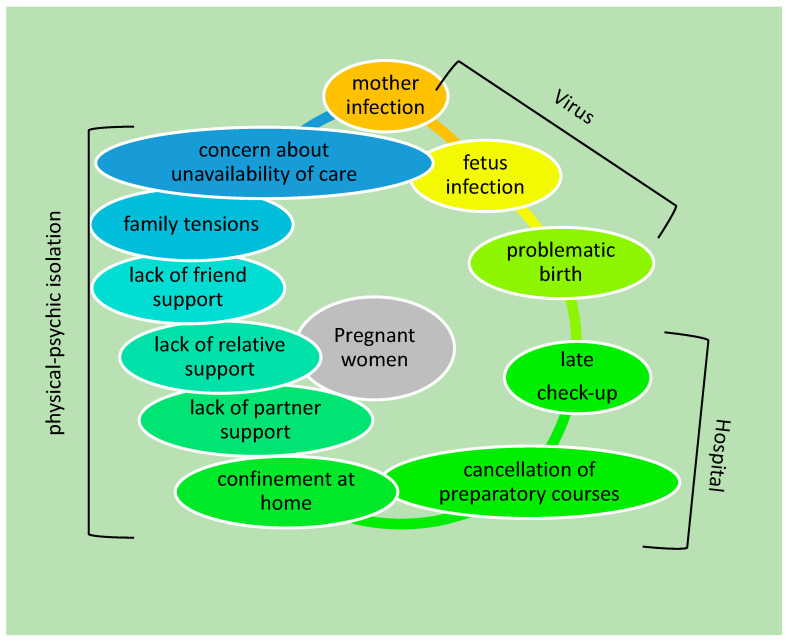
Schematic representation of the main causes affecting the mental health of pregnant women during the COVID-19 pandemic.

**Figure 2 ijerph-19-11497-f002:**
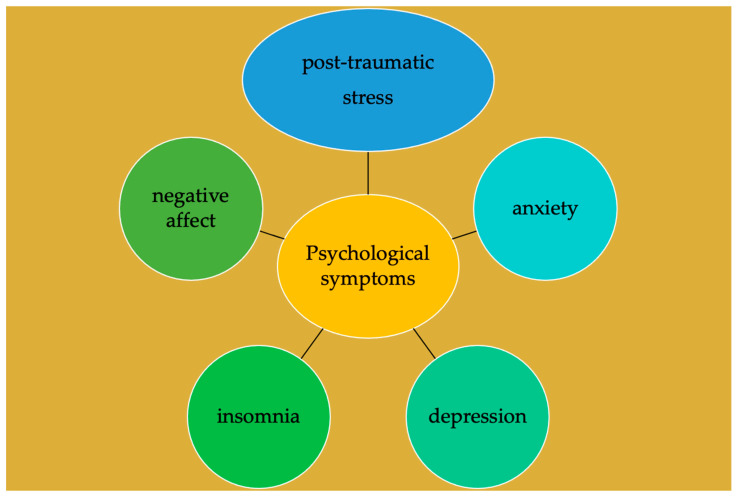
Schematic representation of the main mental disorders in pregnant women during the COVID-19 pandemic induced by causes reported in Figure 1.

**Figure 3 ijerph-19-11497-f003:**
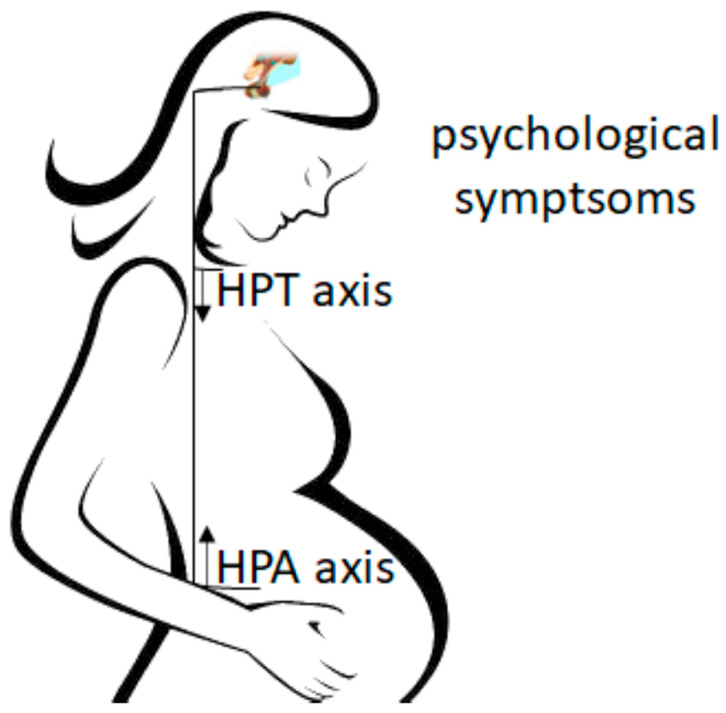
Hypothesis of the involvement of the hypothalamic–pituitary–adrenal axis (HPA) and the hypothalamic–pituitary–thyroid axis (HPT) in the mental health of pregnant women during the COVID-19 pandemic. The viral infection is responsible for the release of proinflammatory cytokines that cross the blood–brain barrier and activate the neuroendocrine system, especially HPA. Moreover, HPA is activated by stress and HPT is inhibited by stress. The deregulation of both HPA and HPT affects mental health with consequent psychological symptoms.

## Data Availability

Not applicable.

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
