# Peer review of "Possible Stress–Neuroendocrine System–Psychological Symptoms Relationship in Pregnant Women during the COVID-19 Pandemic"

_ijerph, 2022, doi:10.3390/ijerph191811497_

Round 1

Reviewer 1 Report

This review was interesting and relevant.  The reviews were thorough, well-summarized, and nicely organized.  The hypothesized HPA/HPT disruption from covid flows nicely from the existing literature.  Implications and suggestions are practical, with the potential to help women and babies adjust to a crucial time of life. 

A few minor grammatical errors/typos in the manuscript should be caught via final proofing.

Author Response

Comments and Suggestions for Authors

This review was interesting and relevant.  The reviews were thorough, well-summarized, and nicely organized.  The hypothesized HPA/HPT disruption from covid flows nicely from the existing literature.  Implications and suggestions are practical, with the potential to help women and babies adjust to a crucial time of life. 

A few minor grammatical errors/typos in the manuscript should be caught via final proofing.

Thank you very much! Minor grammatical errors/typos have been corrected

Reviewer 2 Report

This is a good literature review. Few comments are listed in the following.

1. It would be good to add "possible relationships" or "possible connections between" in your topic since you mentioned the long-term effects are still not defined in your conclusion.

2. Having stress and related symptoms during pregnancy can be caused by different factors, not only due to COVID 19.  It would be good to mention it in the literature review.

3. You have compared stress and related symptoms before and after the COVID pandemic among pregnant women. While your literature findings showed there might have some significant differences, you may forget to check if demographic factors may also have influences on this issue.

Those demographic factors may be having a lower income caused by the COVID pandemic, unmarried women, or lost relatives due to the COVID pandemic.

4. You have reviewed many references mainly from European countries, two North American countries, and one Asian country which is China.

Maybe you can categorize them into different regions under each topic. It would be better to help readers obtain information more quickly.

5. Can you add a short limitation section?

Author Response

Comments and Suggestions for Authors

This is a good literature review. Few comments are listed in the following.

  1. It would be good to add "possible relationships" or "possible connections between" in your topic since you mentioned the long-term effects are still not defined in your conclusion.

Right! It has been included in the title.

  1. Having stress and related symptoms during pregnancy can be caused by different factors, not only due to COVID 19.  It would be good to mention it in the literature review.

It has been reported (lines 348-261).

  1. You have compared stress and related symptoms before and after the COVID pandemic among pregnant women. While your literature findings showed there might have some significant differences, you may forget to check if demographic factors may also have influences on this issue.

It has been reported (lines 361-363).

  1. Those demographic factors may be having a lower income caused by the COVID pandemic, unmarried women, or lost relatives due to the COVID pandemic.

It has been reported (lines 363-365).

  1. You have reviewed many references mainly from European countries, two North American countries, and one Asian country which is China.

Maybe you can categorize them into different regions under each topic. It would be better to help readers obtain information more quickly.

It has been performed  in 3.2.1. and 3.2.2.

  1. Can you add a short limitation section?

Limitation section has been included (lines 510-517).

Reviewer 3 Report

The authors address a relevant topic and show they conducted a significant review of recent publications about maternal mental health and COVID-19. Unfortunately, the writing style makes it hard to follow the authors ‘argument. The ideas and findings from the several revised articles are listed but not integrated. Given the extensive review that was conducted, it would be great to present the ideas as a cohesive and coherent reflection, which would certainly contribute to the field.  Also, there are several grammar mistakes throughout the manuscript, which could be corrected. Maybe having assistance from a native English speaker, or professional editing services could be useful. Please find below specific comments.

Page 1. What do you mean by “physiological but critical period for women”. I do not think that a period of time can be described as physiological, although and event may be labelled as such. Also, “but” is commonly used when addressing two opposite or contradictory conditions, I may be wrong, but physiological and critical do not oppose each other. Actually, physiological events – such as medical conditions and disease – can be very critical. I would appreciate if you could consider to rephrase this initial sentence.  

Line 40 “the prolactin production is responsible for the low anxiety levels in lactating women”. There is no doubt that prolactin positively contributes to maternal emotional wellbeing. That said, many lactating women, especially those who experiences difficulties with breastfeeding, experience anxiety.

“Thus, hormonal modifications allow the pregnancy to proceed in a good state of physical-psychic balance and maternal well-being includes also having a positive childbirth experience”. I would be cautious with this expression. Although pregnancy and childbirth are often considered joyous experiences, a significant number of women experience symptoms of depression and anxiety during pregnancy, which are a risk factor for postnatal depression. Hormonal changes may contribute to emotional wellbeing, but there are many, very diverse factors underlying maternal mental health and childbirth experience. Obviously, childbirth is triggered by hormonal changes, yet variables associated with the health providers, hospital, support available to the woman, previous childbirth experiences, and so many others, also play a significant role in how childbirth is experienced.

Page 2, lines 73-76. You use the expression COVID-19 and Covid 19. Both are correct, please be consistent and use one throughout the manuscript.

Page 3, lines 92-96. It reads “It should also be considered that percentage of cesarean delivery was increased and breastfeeding was reduced in women exposed to COVID-19 compared to control women. So at the moment the studies on the physical consequences, in terms of direct damage and complications induced by COVID-19 in pregnant women, are conflicting”. It would be interesting to know if the increased rates of cesarean sections was associated with medical complications of the women or because hospitals and health providers were overwhelmed and focused on the COVID crisis, which lead them to schedule cesarean sections to limit the involvement of the staff in  medical areas other than those related to respiratory problems. Also, the low levels of breastfeeding may be associated with the high rates of cesarean sections, or  mother-infant separation, limited support available to women who have difficulties with breastfeeding, or maternal mental health difficulties, among others. I do not know if these consequences are directly or indirectly caused by COVID-19. I would appreciate if you could refine your argument.

Page 3, lines 100-102 it reads “A difficult pregnancy and/or birth leaded 100 to the onset of postpartum depression that typically occurs during the end of gestation 101 until the end of the first year of infant's life [22]”. By definition, postpartum depression presents in the postpartum period and not during gestation. If symptoms of depression occur during pregnancy, they are considered antenatal depression, and they are often a risk factor for postnatal depression.

Line 103 it reads “Lebel et al [23] reported that the symptoms 102 of anxiety and depression were found in 10%-25% of pregnancy women in normal 103 conditions”. What are “normal conditions”? Do you mean a community sample? Or a non-clinical population? Please clarify.

Page 3, lines 121-123. Typically, when presenting an idea authors include references that support their claim. If you present contradictory evidence (i.e., the Mount Sinai Hospital study) it is helpful to the reader to explain the purpose of including it. Otherwise, it can be included in the discussion. Whatever you decide, it is important to either use references to support your argument, or provide possible explanations for contradictory evidence.

Page 5, lines 185. Why do you mention the increase in alcohol consumption? The paragraph addressed the relevance of social support in maternal mental health. Did alcohol consumption increase in perinatal women or in their support individuals? Did alcohol consumption negatively impacted the support women received? The sentence is not integrated with the rest of the ideas presented.

Section 3.2.1 Covid 19 period. You present several studies conducted with perinatal women. As a reader, is hard to follow your argument, the studies and data are listed, but no presented in a cohesive way. Please consider revising your writing. The same applies for sections 3.2.2 and 4. As a reader I am interested in a reflection based on the integration of findings from different studies, not in a summary on individual findings.

Page 8, lines 285-286. Is there an explanation to why women during the pandemic had more depression but lower stress than women assessed before the COVID crisis? It would be interesting to know.

Conclusion.

You have not mentioned the long-term impact of maternal stress on the unborn baby until this section. This is a very relevant topic, but it is not advisable to introduce new ideas in the conclusion, this should be oriented to wrapping up the arguments presented earlier. You may want to present this earlier or as a suggestion for future studies and possible implications from this review.

Line 373. What do you mean by “innovative information aimed at a pandemic”. The expression is not clear.

Line 377. What do you mean by “family drama”? Is not a formal expression to be included in an article.

Lines 378. How do you prevent discomfort from the pandemic? Giving concrete suggestions could be a good idea.

Lines 382. You mention that women could “resume a normal life”. “Normal” is a tricky expression, and after having a baby, life is certainly not normal (AKA as it was before). Please use another expression.

Figure 1. I think the text in the grey bubble should read “pregnant women” instead of “pregnancy women”

Figure 2. Is labelled as Figure 1 instead of Figure 2. Also, what do the colorful circles with no text mean? Other minor disorders? Are they decorative? If they do not have a communicative purpose, it would be better to remove them.

Author Response

Comments and Suggestions for Authors

The authors address a relevant topic and show they conducted a significant review of recent publications about maternal mental health and COVID-19. Unfortunately, the writing style makes it hard to follow the authors ‘argument. The ideas and findings from the several revised articles are listed but not integrated. Given the extensive review that was conducted, it would be great to present the ideas as a cohesive and coherent reflection, which would certainly contribute to the field.  Also, there are several grammar mistakes throughout the manuscript, which could be corrected. Maybe having assistance from a native English speaker, or professional editing services could be useful. Please find below specific comments.

Page 1. What do you mean by “physiological but critical period for women”. I do not think that a period of time can be described as physiological, although and event may be labelled as such. Also, “but” is commonly used when addressing two opposite or contradictory conditions, I may be wrong, but physiological and critical do not oppose each other. Actually, physiological events – such as medical conditions and disease – can be very critical. I would appreciate if you could consider to rephrase this initial sentence.  

Thanks for this remark. Indeed it is assumed that it is a physiological period and rightly it is not in opposition to critical. Physiological was removed.

Line 40 “the prolactin production is responsible for the low anxiety levels in lactating women”. There is no doubt that prolactin positively contributes to maternal emotional wellbeing. That said, many lactating women, especially those who experiences difficulties with breastfeeding, experience anxiety.

It has been considered (lines 41-43).

“Thus, hormonal modifications allow the pregnancy to proceed in a good state of physical-psychic balance and maternal well-being includes also having a positive childbirth experience”. I would be cautious with this expression. Although pregnancy and childbirth are often considered joyous experiences, a significant number of women experience symptoms of depression and anxiety during pregnancy, which are a risk factor for postnatal depression. Hormonal changes may contribute to emotional wellbeing, but there are many, very diverse factors underlying maternal mental health and childbirth experience. Obviously, childbirth is triggered by hormonal changes, yet variables associated with the health providers, hospital, support available to the woman, previous childbirth experiences, and so many others, also play a significant role in how childbirth is experienced.

Thanks again, it has been included in the text (Lines 46-54).

Page 2, lines 73-76. You use the expression COVID-19 and Covid 19. Both are correct, please be consistent and use one throughout the manuscript.

It has been corrected along the text.

Page 3, lines 92-96. It reads “It should also be considered that percentage of cesarean delivery was increased and breastfeeding was reduced in women exposed to COVID-19 compared to control women. So at the moment the studies on the physical consequences, in terms of direct damage and complications induced by COVID-19 in pregnant women, are conflicting”. It would be interesting to know if the increased rates of cesarean sections was associated with medical complications of the women or because hospitals and health providers were overwhelmed and focused on the COVID crisis, which lead them to schedule cesarean sections to limit the involvement of the staff in  medical areas other than those related to respiratory problems. Also, the low levels of breastfeeding may be associated with the high rates of cesarean sections, or  mother-infant separation, limited support available to women who have difficulties with breastfeeding, or maternal mental health difficulties, among others. I do not know if these consequences are directly or indirectly caused by COVID-19. I would appreciate if you could refine your argument.

It has been included (lines 118-122).

Page 3, lines 100-102 it reads “A difficult pregnancy and/or birth leaded 100 to the onset of postpartum depression that typically occurs during the end of gestation 101 until the end of the first year of infant's life [22]”. By definition, postpartum depression presents in the postpartum period and not during gestation. If symptoms of depression occur during pregnancy, they are considered antenatal depression, and they are often a risk factor for postnatal depression.

It has been corrected (Line 132).

Line 103 it reads “Lebel et al [23] reported that the symptoms 102 of anxiety and depression were found in 10%-25% of pregnancy women in normal 103 conditions”. What are “normal conditions”? Do you mean a community sample? Or a non-clinical population? Please clarify.

It has been clarified (line 134).

Page 3, lines 121-123. Typically, when presenting an idea authors include references that support their claim. If you present contradictory evidence (i.e., the Mount Sinai Hospital study) it is helpful to the reader to explain the purpose of including it. Otherwise, it can be included in the discussion. Whatever you decide, it is important to either use references to support your argument, or provide possible explanations for contradictory evidence.

It has been rewritten (lines 157-161).

Page 5, lines 185. Why do you mention the increase in alcohol consumption? The paragraph addressed the relevance of social support in maternal mental health. Did alcohol consumption increase in perinatal women or in their support individuals? Did alcohol consumption negatively impacted the support women received? The sentence is not integrated with the rest of the ideas presented.

It has been rewritten (lines 227-231).

Section 3.2.1 Covid 19 period. You present several studies conducted with perinatal women. As a reader, is hard to follow your argument, the studies and data are listed, but no presented in a cohesive way. Please consider revising your writing. The same applies for sections 3.2.2 and 4. As a reader I am interested in a reflection based on the integration of findings from different studies, not in a summary on individual findings.

Sections 3.2.1, 3.2.2. and 4 were rewritten according your suggestions.

Page 8, lines 285-286. Is there an explanation to why women during the pandemic had more depression but lower stress than women assessed before the COVID crisis? It would be interesting to know.

Information has been included (lines 424-427).

Conclusion.

You have not mentioned the long-term impact of maternal stress on the unborn baby until this section. This is a very relevant topic, but it is not advisable to introduce new ideas in the conclusion, this should be oriented to wrapping up the arguments presented earlier. You may want to present this earlier or as a suggestion for future studies and possible implications from this review.

It has been included in “study limitations” (lines 512-515).

Line 373. What do you mean by “innovative information aimed at a pandemic”. The expression is not clear.

It has been corrected (line 541-542).

Line 377. What do you mean by “family drama”? Is not a formal expression to be included in an article.

It has been corrected (lines 545,546).

Lines 378. How do you prevent discomfort from the pandemic? Giving concrete suggestions could be a good idea.

It has been included (lines 548-550).

Lines 382. You mention that women could “resume a normal life”. “Normal” is a tricky expression, and after having a baby, life is certainly not normal (AKA as it was before). Please use another expression.

 The expression has been corrected (line 553).

Figure 1. I think the text in the grey bubble should read “pregnant women” instead of “pregnancy women”

Figure 1 has been corrected.

Figure 2. Is labelled as Figure 1 instead of Figure 2. Also, what do the colorful circles with no text mean? Other minor disorders? Are they decorative? If they do not have a communicative purpose, it would be better to remove them.

Figure 1 has been changed.

Reviewer 4 Report

I would like to suggest you rearrange the text of the manuscript. The objective of the study was to describe the relationship between stress, neuroendocrine system, and psychological symptoms in pregnant women during the COVID-19 pandemic.

Accordingly, the manuscript needs to involve the following parts.

1)      Common immunological changes in pregnancy. During pregnancy, we can see the Th1 inflammation‐like condition early in pregnancy, with the shift to a temporal Th2 biased immune tolerance state during the second trimester and a second shift during pregnancy (Sarapultsev A, Sarapultsev P. Changes in the immune environment during pregnancy may affect the risk of developing severe complications in patients with COVID-19. Am J Reprod Immunol. 2020 Sep;84(3):e13285. doi: 10.1111/aji.13285.). These shifts can significantly affect the onset of depression symptoms or even lead to them. /// 2. Viral infection and immune system in pregnancy

2)      Common immunological changes in pregnancy during COVID-19 (Misra SS, Ahirwar AK, Sakarde A, Kaim K, Ahirwar P, Jahid M, Sorte SR, Lokhande SL, Kaur AP, Kumawat R. COVID-19 infection in pregnancy: a review of existing knowledge. Horm Mol Biol Clin Investig. 2022 Feb 16. doi: 10.1515/hmbci-2021-0081)

3)      The prevalence of stress-related disorders during epidemic in society and among pregnant women during other viral infections (if there is data).

4)      The prevalence of stress-related disorders during epidemic in society and among pregnant women during COVID-19 (if there is data) [Luo Y, Zhang K, Huang M, Qiu C. Risk factors for depression and anxiety in pregnant women during the COVID-19 pandemic: Evidence from meta-analysis. PLoS One. 2022 Mar 4;17(3):e0265021. doi: 10.1371/journal.pone.0265021.] // COVID-19 period versus pre-COVID-19 period

5)      The factors which are unique for COVID-19 (mostly socio-psychological ones as occupational stress, low income due to restrictions and government policy, lockdowns and changes in medical practices….) /// 3. The psychological impact of COVID-19 in pregnancy

6)      Discussion and future perspectives.

Other remarks.

·         Chapter 4. Possible involvement of the hypothalamic-pituitary-adrenal axis and hypothalamic- 299

·         pituitary-thyroid axis in the mental health impact of COVID-19. There are no data on the direct effect of SARS on the hypothalamic-pituitary-adrenal axis and the hypothalamic-pituitary-thyroid axis. These axes are involved in the stress response of any etiology.  If you have some information on the direct effects of the virus, please focus on it. If not, please delete this part.

·         Figures. All figures are marked as Figure 1. The quality is low. Please give other figures.

·         The reference list should include at least the studies mentioned above and should be enriched with the meta-analyses and recent Reviews on that topic.

Author Response

Comments and Suggestions for Authors

I would like to suggest you rearrange the text of the manuscript. The objective of the study was to describe the relationship between stress, neuroendocrine system, and psychological symptoms in pregnant women during the COVID-19 pandemic.

Accordingly, the manuscript needs to involve the following parts.

1)      Common immunological changes in pregnancy. During pregnancy, we can see the Th1 inflammation‐like condition early in pregnancy, with the shift to a temporal Th2 biased immune tolerance state during the second trimester and a second shift during pregnancy (Sarapultsev A, Sarapultsev P. Changes in the immune environment during pregnancy may affect the risk of developing severe complications in patients with COVID-19. Am J Reprod Immunol. 2020 Sep;84(3):e13285. doi: 10.1111/aji.13285.). These shifts can significantly affect the onset of depression symptoms or even lead to them. /// 2. Viral infection and immune system in pregnancy

The articles have been included (lines 78-81, 111-113].

2)      Common immunological changes in pregnancy during COVID-19 (Misra SS, Ahirwar AK, Sakarde A, Kaim K, Ahirwar P, Jahid M, Sorte SR, Lokhande SL, Kaur AP, Kumawat R. COVID-19 infection in pregnancy: a review of existing knowledge. Horm Mol Biol Clin Investig. 2022 Feb 16. doi: 10.1515/hmbci-2021-0081)

The article has been included (lines 95-98].

3)      The prevalence of stress-related disorders during epidemic in society and among pregnant women during other viral infections (if there is data).

4)      The prevalence of stress-related disorders during epidemic in society and among pregnant women during COVID-19 (if there is data) [Luo Y, Zhang K, Huang M, Qiu C. Risk factors for depression and anxiety in pregnant women during the COVID-19 pandemic: Evidence from meta-analysis. PLoS One. 2022 Mar 4;17(3):e0265021. doi: 10.1371/journal.pone.0265021.] // COVID-19 period versus pre-COVID-19 period

5)      The factors which are unique for COVID-19 (mostly socio-psychological ones as occupational stress, low income due to restrictions and government policy, lockdowns and changes in medical practices….) /// 3. The psychological impact of COVID-19 in pregnancy

All articles and information requested in 3-5 have been reported in the chapter “COVID-19 period” and in “COVID-19 period versus non-COVID-19 period.

6)      Discussion and future perspectives.

Discussion and future perspectives were included from Conclusions with changes of the chapter.

Other remarks.

  • Chapter 4. Possible involvement of the hypothalamic-pituitary-adrenal axis and hypothalamic- 299
  • pituitary-thyroid axis in the mental health impact of COVID-19. There are no data on the direct effect of SARS on the hypothalamic-pituitary-adrenal axis and the hypothalamic-pituitary-thyroid axis. These axes are involved in the stress response of any etiology.  If you have some information on the direct effects of the virus, please focus on it. If not, please delete this part.

The studies are very few and are present in the text (lines 469-473). This chapter wants to be an original part of stimulus for future studies. It has been included in the “Discussion and future perspectives”. We would like to leave it.

  • Figures. All figures are marked as Figure 1. The quality is low. Please give other figures.

The figures have been improved.

  • The reference list should include at least the studies mentioned above and should be enriched with the meta-analyses and recent Reviews on that topic.

All text has been revised with inclusion of suggested references and many others.

Round 2

Reviewer 3 Report

I appreciate the significant changes you have made to the manuscript. I have minor comments (please see below) and maintain my original suggestion to revise the English writing. Although it is improved, there are still several minor writing mistakes in the text. 

Page 2, line 42 “even if breastfeeding women can cause anxiety especially in 42 women who have difficulty with breastfeeding.” The writing in unclear, please revise.

Lines 46-53. I appreciate that you included my suggestion. Please rephrase it so it is more formal, and not a copy of my comment.

Line 218 The difficulty of interpersonal relationships, studied by Social Support Ef- 217 fectiveness Questionnaire (SSEQ), had been highlighted by the Canadian study 218 reported above [27]”. In the previous paragraph two Canadian studies are mentioned, please clarify which one are you referring to.

Please be consistent in the use of COVID or Covid throughout the text.

Author Response

I appreciate the significant changes you have made to the manuscript. I have minor comments (please see below) and maintain my original suggestion to revise the English writing. Although it is improved, there are still several minor writing mistakes in the text. 

Thank you, the English language has been revised.

Page 2, line 42 “even if breastfeeding women can cause anxiety especially in 42 women who have difficulty with breastfeeding.” The writing in unclear, please revise.

 The statement has been revised (line 42)

Lines 46-53. I appreciate that you included my suggestion. Please rephrase it so it is more formal, and not a copy of my comment.

The text has been rephrased (lines 61-68)

Line 218 The difficulty of interpersonal relationships, studied by Social Support Ef- 217 fectiveness Questionnaire (SSEQ), had been highlighted by the Canadian study 218 reported above [27]”. In the previous paragraph two Canadian studies are mentioned, please clarify which one are you referring to.

The reference has been reported [27].

Please be consistent in the use of COVID or Covid throughout the text.

Thank you again, COVID was used

Reviewer 4 Report

Dear authors, thank you for the conducted work. The manuscript was improved and I hope will find the readers! 

Author Response

Thank you very much